# Evaluation of In Vitro Production Capabilities of Indole Derivatives by Lactic Acid Bacteria

**DOI:** 10.3390/microorganisms13010150

**Published:** 2025-01-13

**Authors:** Bingyang Ma, Yan Zhao, Liping Liu, Jianguo Xu, Qingping Hu, Saisai Feng, Liangliang Zhang

**Affiliations:** 1Institute of Food Sciences, Shanxi Normal University, Taiyuan 030031, China; mby13273800347@163.com (B.M.); lipingliu_snux@outlook.com (L.L.); xjg71@163.com (J.X.); 2College of Life Science, Shanxi Normal University, Taiyuan 030031, China; hqp72@163.com; 3Jiangsu Key Laboratory of Marine Bioresources and Environment, Jiangsu Ocean University, Lianyungang 222005, China; 18811990165@163.com; 4Jiangsu Key Laboratory of Marine Biotechnology, Jiangsu Ocean University, Lianyungang 222005, China

**Keywords:** lactic acid bacteria, indole-3-carboxaldehyde, indole-3-lactic acid, *Lactiplantibacillus*, species- or genus-dependent capabilities for metabolizing indole derivatives

## Abstract

Lactic acid Bacteria (LAB) convert tryptophan to indole derivatives and induce protective IL-22 production in vivo. However, differences in metabolizing capabilities among LAB species have not been widely investigated. In the present study, we compared the capabilities of 186 LAB strains to produce four kinds of indole derivatives, including indole-3-carboxaldehyde (IAId), indole-3-lactic acid (ILA), indole-3-propanoic acid (IPA), and indole-3-acetic acid (IAA). These strains were isolated from fermented foods, dairy products, and the feces of healthy individuals, as well as from fish and shrimp from Shanxi and Jiangsu provinces. They represent 15 genera, including *Bifidobacterium*, *Enterococcus*, *Lacticaseibacillus*, *Lactiplantibacillus*, *Lactobacillus*, *Lactococcus*, *Limosilactobacillus*, *Pediococcus*, *Streptococcus*, *Weissella*, *Latilactobacillus*, *Levilactobacillus*, *Ligilactobacillus*, and *Loigolactobacillus*. The results indicate widespread IAId-producing capabilities in LAB strains, with positive rates of approximately 90% (106/117) and 100% (69/69) among strains from Shanxi and Jiangsu provinces, respectively. The concentrations of IAId ranged from 72.42 ng/mL to 423.14 ng/mL in all positive strains from Shanxi Province and from 169.39 ng/mL to 503.51 ng/mL in strains from Jiangsu Province. Intriguingly, we also observed specific ILA-producing capabilities in *Lactiplantibacillus* strains, with positive rates of 55.17% (16/29) and 80.95% (17/21) among strains isolated from Shanxi and Jiangsu provinces, respectively. The overall detection rates of ILA among all tested strains (including both *Lactiplantibacillus* and other genus strains) were 17.9% (21/117) and 26.1% (18/69). The concentrations of ILA in positive strains ranged from 12.22 ng/mL to 101.86 ng/mL and from 5.75 ng/mL to 62.96 ng/mL from Shanxi and Jiangsu provinces, respectively. IPA and IAA were not detected in any strains. Finally, these indole derivative-producing capabilities were not related to their geographical origins or isolation sources. The current study provides insights into the species- or genus-dependent capabilities for metabolizing indole derivatives. Defining the specific roles of LAB in indole derivative metabolism will uncover the exact physiological mechanisms and be helpful for functional strain screening.

## 1. Introduction

In recent years, lactic acid bacteria (LAB) have attracted significant attention for their ability to produce a variety of bioactive compounds, particularly indole derivatives, which are linked to a range of beneficial health effects [1,2,3,4,5,6,7,8,9,10,11]. Among these metabolites, indole-3-aldehyde (IAId), indole-3-lactic acid (ILA), indole-3-propanoic acid (IPA), indole-3-acetic acid (IAA), and others play pivotal roles in modulating immune responses, enhancing gut health, and potentially influencing systemic diseases [2,3,4,5,6,7,8,9,10,11,12]. LAB, including well-known probiotics such as *Limosilactobacillus reuteri* and *Lactobacillus acidophilus*, produce these compounds through the metabolism of tryptophan, and their interaction with the aryl hydrocarbon receptor (AhR) has been demonstrated to activate immune pathways, including the secretion of protective cytokines like interleukin-22 (IL-22) [2].

Despite the many investigations into the role of these indole derivatives in immune regulation and gut microbiota balance, much remains unknown about the full metabolic pathways involved, the diversity of LAB strains that can produce these metabolites, and the geographical or taxonomic factors that may influence their production [13,14]. The production of indole derivatives can vary significantly among different LAB strains, potentially depending on their origin and the specific microbial ecosystem they inhabit [15]. While studies have established the presence of these metabolites in LAB strains from various sources, including dairy products, fermented foods, and human feces, the ecological and genetic factors that influence this variation are still not fully understood.

In this study, by systematically examining LAB strains from different geographical regions and ecological niches, we aim to identify potential regional, niche, or taxonomic differences in their ability to produce key indole metabolites, including IAId, IPA, IAA, and ILA. Understanding these variations may reveal new insights into how LAB contribute to host health, particularly in terms of immune modulation, gut barrier function, and inflammation, and also provide broader implications for the development of LAB-based functional probiotics.

## 2. Materials and Methods

### 2.1. Bacteria Strains and Growth Conditions

The LAB strains used in the present study are listed in the Appendix A. Except for *L. reuteri* ATCC 23272, which was purchased from the American Type Culture Collection, part strains were isolated from the fermented products and the feces of healthy individuals from Shanxi Province, China. The other strains were isolated from the feces of various fish and shrimp, fermented or dairy products, and the feces of healthy individuals from Lianyungang City, Jiangsu Province, China.

To determine the in vitro indole derivative-producing capabilities of single or composite strains, resting bacteria were prepared and utilized for UPLC-MS detection following the protocol described by Han et al. [16]. Briefly, 200 μL of freshly prepared bacterial suspensions was centrifuged at 12,000 g for 10 min at 4 °C. The pellets were washed twice with pre-reduced PBS buffer and then resuspended in 2 mL of PBS buffer at room temperature for 45 min. Subsequently, the pellets were centrifuged again, and then resuspended in 1 mL of reaction solution (comprising 2 mM tryptophan and 10 mM α-ketoglutarate in PBS) under anaerobic conditions for 2 h. An anaerobic glove box (BasicI, Jiangxue Company, Chongqing, China), purged with 80% N₂, 10% CO₂, and 10% H₂, was used to maintain anaerobic conditions. Finally, the supernatants of the reaction solutions were collected and utilized for subsequent UPLC-MS analysis. Before injection, the supernatants were passed through a microfiltration membrane (0.22 μm).

### 2.2. 16S rDNA Sequencing and Construction of Phytological Tree

Partial or full-length 16S rDNA of all strains used in this study was amplified using primer pairs 314E/806R or 27F/1492R, and their sequences are listed in Table 1 [17]. The amplification products were then sent to Shanghai Shenggong Bioengineering Co., Ltd. (Shanghai, China), where Sanger sequencing was performed. For the construction of the phytological tree, *Escherichia coli* was used as the outgroup strain, and multiple sequence alignment was performed using MUSCLE [18]. The aligned sequences were then trimmed using trimAl [19]. Finally, maximum likelihood trees were constructed using IQ-TREE in MFP (ModelFinder Plus) mode [20]. A total of 1000 bootstrap replications were performed to obtain consensus trees. Typically, a single tree was constructed for each primer pair.

### 2.3. Determination of Indole Derivatives in the Culture Supernatants by UPLC-MS

Here, four kinds of indole derivatives (IAId, ILA, IPA, and IAA) were determined using an Ultimate 3000 HPLC system (Thermo Scientific) coupled to an ultra-high resolution QTOF mass spectrometer (Bruker Impact II) equipped with an ESI interface. A Waters ACQUITY UPLC HSS T3 Column (1.8 μm, 2.1 mm × 100 mm) was employed for liquid chromatography. A 5 mM sodium formate solution was used to perform mass calibration prior to each run. The mobile phase for liquid chromatography consisted of water with 0.1% (*v*/*v*) formic acid (A) and acetonitrile (B) with a flow rate of 0.3 mL/min at 30 °C. The injection volume was 5 μL. For the gradient elution, the percentage of B was set as follows: from 0 min to 0.5 min, maintained at 0.5% B; from 0.5 min to 1 min, linearly increased from 0.5% to 35%; from 1 min to 4 min, maintained at 35%; from 4 min to 5 min, linearly increased from 35% to 70%; after 0.5 min at 70% B, the percentage of B increased to 100% within 0.5 min; from 6 min to 8 min, maintained at 100%. MS detection was performed in negative ion mode with the following parameters: ESI spray voltage of 2 kV with an end plate offset of −500 V, capillary voltage of 2.5 kV, a nebulizer pressure of 2 bar, a dry gas flow of 4.0 L/min, a heater temperature of 200 °C, and a scan range ranging from 50 to 1000 *m*/*z*. The standard curves for ILA, IAId, IPA, and IAA are shown in Table 2 and were used for quantification. Quality control (QC) samples were prepared by equally mixing each tested sample to monitor the stability of the target compounds throughout the entire determination period. Phosphate-buffered saline (PBS) was used as a negative control. QC and PBS samples were analyzed every 10 samples. Both IAId and ILA exhibited a coefficient of variation (CV%) of QC below 10%.

### 2.4. Replication Design and Statistical Analysis

To verify whether indole derivative production capabilities are influenced by a single culture and detection condition, we randomly selected 20 strains from all the tested strains, including 10 with ILA production capabilities and 10 without. Additionally, two further biosample-level replications were performed. The consensus was that the quantitative results from all three independent experiments were considered reliable. To compare differences among genera and geographical regions, chi-square tests were conducted. To assess whether isolation sources were related to indole derivative production capabilities, a Fisher’s exact test was performed.

## 3. Results

### In Vitro Production Capabilities of Indole Derivatives by Lactic Acid Bacteria

To investigate the genus- or species-dependent in vitro indole derivative-producing abilities of LAB, we initially determined the in vitro production of IAId, ILA, IPA, and IAA by 117 LAB strains isolated from fermented products or feces of healthy individuals in Shanxi Province, China. The results are presented in Table 3 and Appendix A. These strains belonged to 11 genera: *Bifidobacterium*, *Enterococcus*, *Lacticaseibacillus*, *Lactiplantibacillus*, *Lactobacillus*, *Lactococcus*, *Limosilactobacillus*, *Pediococcus*, *Streptococcus*, and *Weissella*. None of these strains produced IAA or IPA. Approximately 90% of the strains produced IAId, ranging from 72.42 ng/mL to 423.14 ng/mL. ILA was only detected in strains from four genera: *Lactiplantibacillus*, *Enterococcus*, *Weissella*, and *Pediococcus*, with positive rates of 55.17% (16/29), 6.90% (2/29), 5.88% (1/17), and 15.38% (2/13), respectively. *Lactiplantibacillus* strains exhibited notably higher ILA-producing frequencies, with levels ranging from 12.22 ng/mL to 101.86 ng/mL.

To assess the influence of geographical origin, we also determined the indole derivative-producing capabilities of 69 LAB strains isolated from the feces of fish or shrimp, fermented or dairy products, and feces of healthy individuals collected in Lianyungang City, Jiangsu Province, China. The results are presented in Table 4 and Appendix A. These strains included nine genera: *Lacticaseibacillus*, *Lactiplantibacillus*, *Limosilactobacillus*, *Pediococcus*, and *Weissella* and the previously untested genera *Latilactobacillus*, *Levilactobacillus*, *Ligilactobacillus*, and *Loigolactobacillus*. Again, no IPA or IAA was detected in any of the tested strains. All strains produced IAId, and the positive rate of ILA in *Lactiplantibacillus* strains was 80.95% (17/21). The concentrations of IAId and ILA in these strains were comparable to those observed in the previous batch.

Afterward, we conducted a verification experiment to avoid any incidental observations that might have been caused by a single culture or detection method. We randomly selected 20 strains from all tested strains, including 10 strains with ILA-producing capabilities and 10 strains without, and performed two additional biological replicates. The results show consistent and reliable findings.

Figure 1 presents a heatmap of the production capabilities for four types of indole derivatives across all 186 tested strains, along with their taxonomic classification, geographical origin, and isolation source information. Figure 2 illustrates the combined positive rate across genera and the ranges of ILA and IAId concentrations among the positive samples. As indicated, we observed widespread IAId-producing capabilities among LAB strains and specific ILA-producing capabilities in *Lactiplantibacillus* strains (χ² = 18.958, *p* = 1.336e-05). A key question of interest is whether the specific ILA-producing capabilities of *Lactiplantibacillus* strains are related to geographical factors (Shanxi or Jiangsu) or isolation sources (gut-associated or others). To address this, we performed a chi-square test for geographical origin and a Fisher’s exact test for isolation sources. The results show that ILA-producing capabilities were not related to geographical origins (χ^2^ = 1.2784, *p* = 0.2582) or isolation sources (*p* = 0.3343).

## 4. Discussion

Our research demonstrates that *Lactiplantibacillus* strains exhibit specific capabilities for the production of ILA. In contrast, widespread production of IAId was observed among various LAB strains, regardless of their geographical origin or the source from which they were isolated. This suggests a common trait among LAB strains for producing IAId but highlights the variability in ILA production across different genera and strains.

Interestingly, our results reveal a significant discrepancy when compared to previous studies. *Lactobacillus reuteri* strains, which were expected to produce ILA based on earlier findings, failed to produce ILA in our experimental setup. For instance, Pan et al. had previously reported that several Lactobacillus species, including *L. acidophilus*, *L. fermentum*, *L. reuteri*, and *L. rhamnosus*, were capable of producing indole derivatives such as indole-3-propionic acid (IPyA), ILA, and IAA [14]. However, Pan et al. did not explore the specific production of IAId, which is essential to understanding the broader spectrum of indole-based compounds [15].

Similarly, in a study by Cervantes-Barragan et al., it was reported that *L. reuteri* was one of the primary LAB strains capable of producing ILA, which contributed to the induction of gut intraepithelial CD4+CD8αα+ T cells [4]. However, our findings starkly contrast with this, as *L. reuteri* strains in our study showed no production of ILA at all. This raises important questions regarding the environmental or experimental conditions that could influence the ability of *L. reuteri* to produce ILA.

A possible explanation for these discrepancies could lie in the presence of specific genetic factors required for ILA production. Previous studies have suggested that a gene encoding D-2-hydroxyacid dehydrogenase (ldhA) might be essential for the synthesis of ILA [2]. This gene could be absent, mutated, or regulated differently in the *L. reuteri* strains tested in our study, potentially explaining the lack of ILA production.

Moreover, it is well established that *L. reuteri* and *L. acidophilus* are capable of producing IAId both in culture supernatants and in vivo, suggesting the potential widespread capability of LAB to produce IAId in LAB under various culture conditions, strain differences, and environmental factors [2].

The production of indole-3-propionic acid (IPA) was also explored in this study, with our findings aligning with previous reports that most *Lactobacillus* strains are typically not capable of producing IPA. This result is consistent with earlier studies suggesting that Clostridial species—such as *Clostridium sporogenes*, *Clostridium botulinum*, *Clostridium cadaveris*, and *Peptostreptococcus anaerobius*—are the primary producers of IPA in the gut microbiota [14].

Strengths of this study include its comprehensive examination of various LAB strains, highlighting the variability in indole derivative production across different genera and strains. The investigation of specific compounds like ILA and IAId expands the understanding of LAB metabolic capabilities, offering new insights into their role in gut microbiota. However, some limitations should be acknowledged. The study did not include all indole derivatives, which could have provided a more complete picture of the metabolic profiles of LAB strains. Additionally, the selection of strains based on geographical and source diversity could be more representative to capture a broader scope of LAB strains. Further research is needed to explore the underlying genetic and environmental factors influencing ILA production, especially the discrepancies observed with *Lactobacillus reuteri*.

## 5. Conclusions

According to the results of 186 LAB strains isolated from fermented foods, dairy products, or the feces of healthy individuals, as well as from fish and shrimp from Jiangsu and Shanxi provinces, LAB strains exhibit widespread IAId-producing capabilities. The positive rates were approximately 90% (106/117) and 100% (69/69) among strains from Shanxi and Jiangsu provinces, respectively. Among the 15 tested genera, including *Bifidobacterium*, *Enterococcus, Lacticaseibacillus, Lactiplantibacillus*, *Lactobacillus*, *Lactococcus*, *Limosilactobacillus*, *Pediococcus, Streptococcus, Weissella, Latilactobacillus, Levilactobacillus, Ligilactobacillus*, and *Loigolactobacillus*, *Lactiplantibacillus* most frequently secreted ILA. The positive rates for ILA among strains isolated from the two regions were 55.17% (16/29) and 80.95% (17/21). However, the overall detection rates of ILA among all tested strains (including both *Lactiplantibacillus* and other genus strains) were 17.9% (21/117) and 26.9% (18/67). IPA and IAA were not detected in any of the strains. Finally, these indole derivative-producing capabilities were not found to be related to the geographical origins or isolation sources of the strains. The current study provides insights into the species- or genus-dependent capabilities for metabolizing indole derivatives. Defining the specific roles of LAB in indole derivative metabolism will uncover the exact physiological mechanisms and be helpful for functional strain screening.

## Figures and Tables

**Figure 1 microorganisms-13-00150-f001:**
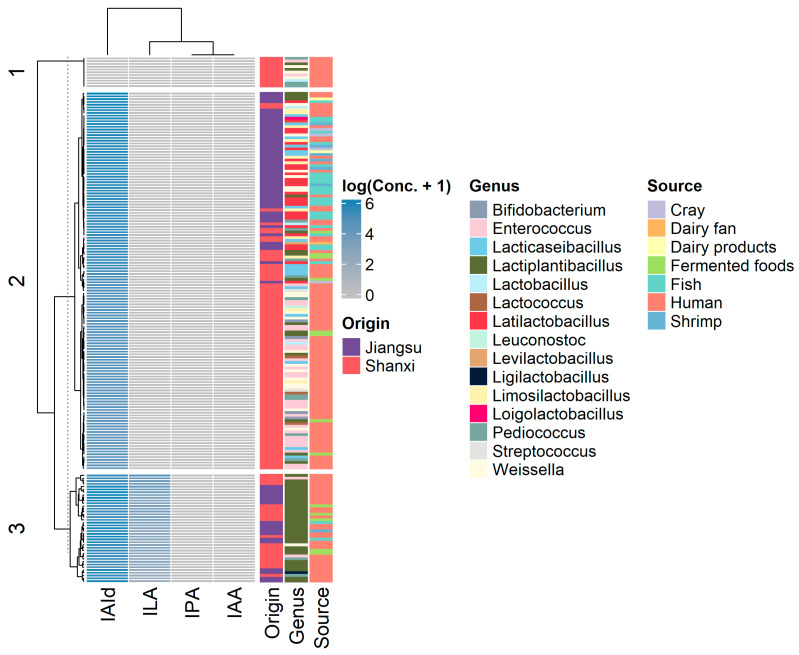
Heatmap displaying the production capabilities of four indole derivatives across 186 LAB strains, with information on their taxonomic classification, geographical origin (Shanxi or Jiangsu), and isolation source (gut-associated or others). Concentration data are presented as the log2 (raw concentration + 1) to avoid zero values and to scale the data for better statistical analysis and visualization.

**Figure 2 microorganisms-13-00150-f002:**
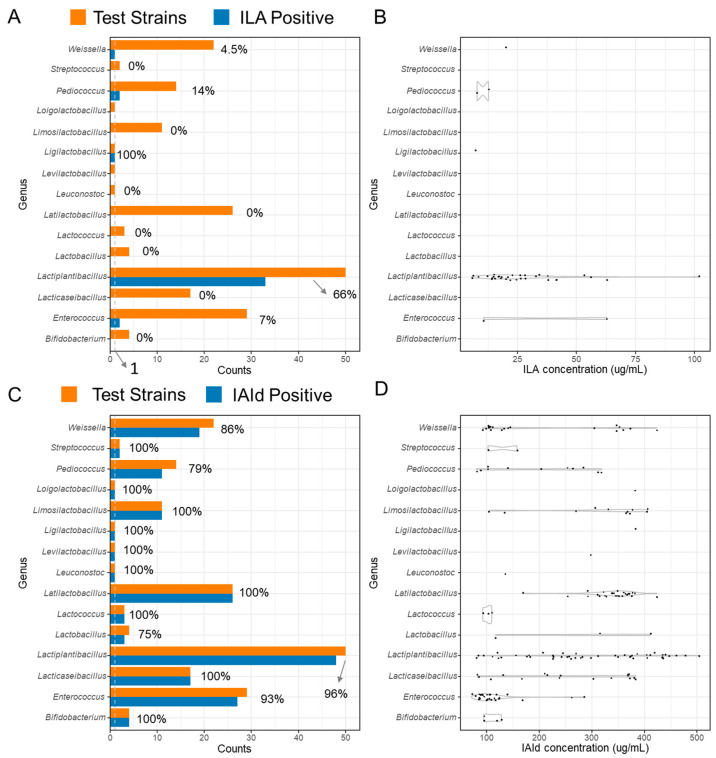
Summary of positive rate across genera and the ranges of ILA-positive and IAId-positive concentrations. *Lactiplantibacillus* showed specific ILA-producing capabilities compared to the other tested genera (chi-square test, χ² = **18.958**, *p* = 1.336e-05).

**Table 1 microorganisms-13-00150-t001:** Primer pairs used for PCR amplification.

ID	Sequence (5′-3′)	Target Gene and Size	Tm
27F	AGAGTTTGATCCTG GCTCA	~1500 bp 16S rDNA	55 °C
1492R	GGTTACCTTGTTACGACTT		
314F	CCTAYGGGRBGCASCAG	~450 bp 16S rDNA V3-V4	58 °C
806R	GGACTACNNGGGTATCTAAT		

**Table 2 microorganisms-13-00150-t002:** *m*/*z*, standard curve, R2, LOD, and LOQ of HPLC-MS detection for ILA, IAId, IPA, and IAA.

Analytes	*m*/*z*	Standard Curve	R ^2^	LOD (ng/mL)	LOQ (ng/mL)
ILA	204.0655	Y = 11.011X − 29573	0.9998	1.5	5
IAId	144.0443	Y = 22164e ^0.0085X^	0.9984	0.5	2.9
IPA	188.0706	Y = 1610.1X − 10960	0.9974	1.3	5.5
IAA	174.0549	Y = 12.405X − 10.562	0.9998	2.4	7.2

**Table 3 microorganisms-13-00150-t003:** Indole derivative-producing capabilities of LAB strains (Shanxi Province) at a genus level.

Genus	No. of Strains	No. of Positive Strains		IAId (ng/mL)		ILA (ng/mL)	
		IAId	ILA	Range	Mean ± SD	Range	Mean ± SD
*Bifidobacterium*	4	4	0	94.74–128.44	109.64 ± 17.08	-	-
*Enterococcus*	29	27	2	72.42–285.3	117.77 ± 49.64	10.6–62.79	36.7 ± 36.91
*Lacticaseibacillus*	10	10	0	82.00–302.78	177.95 ± 74.86	-	-
*Lactiplantibacillus*	29	27	16	80.99–411.07	210.98 ± 80.58	12.22–101.86	33.04 ± 22.8
*Lactobacillus*	4	3	0	117.02–411.59	281.2 ± 150.17	-	-
*Lactococcus*	3	3	0	93.03–109.92	101.98 ± 8.49	-	-
*Leuconostoc*	1	1	0	135.32	135.32	-	-
*Limosilactobacillus*	5	5	0	104.21–330.9	228.84 ± 103.25	-	-
*Pediococcus*	13	10	2	81.28–311.56	183.07 ± 89.77	7.8–12.74	10.27 ± 3.49
*Streptococcus*	2	2	0	103.16–157.95	130.56 ± 38.74	-	-
*Weissella*	17	14	1	92.55–423.14	150.28 ± 94.77	20	20

**Table 4 microorganisms-13-00150-t004:** Indole derivative-producing capabilities of LAB strains (Jiangsu Province) at a genus level.

Genus	No. of Strains	No. of Positive Strains		IAId (ng/mL)		ILA (ng/mL)	
		IAId	ILA	Range	Mean ± SD	Range	Mean ± SD
*Lacticaseibacillus*	7	7	0	337.08–382.89	368.48 ± 14.96	-	-
*Lactiplantibacillus*	21	21	17	343.81–503.51	409.92 ± 43.52	5.75–62.96	22.52 ± 14.24
*Latilactobacillus*	26	26	0	169.39–423.64	336.97 ± 48.41	-	-
*Levilactobacillus*	1	1	0	297.68	297.68	-	-
*Ligilactobacillus*	1	1	1	383.15	383.15	7.1243	7.1243
*Limosilactobacillus*	6	6	0	365.02–405.83	382.03 ± 18.49	-	-
*Loigolactobacillus*	1	1	0	381.96	381.96	-	-
*Pediococcus*	1	1	0	317.98	317.98	-	-
*Weissella*	5	5	0	346.45–372.6	355.42 ± 10.93	-	-

## Data Availability

Data is contained within the article or Appendix A.

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
