# Peer review of "Evaluation of In Vitro Production Capabilities of Indole Derivatives by Lactic Acid Bacteria"

_microorganisms, 2025, doi:10.3390/microorganisms13010150_

Round 1
Reviewer 1 Report
Comments and Suggestions for Authors
Title: Evaluation of In Vitro Production Capabilities of Indole Derivatives by Lactic Acid Bacteria
General comments:
- The manuscript addresses a relevant topic by investigating the production of indole derivatives by LAB strains, an area with significant implications for functional probiotics and health. However, the novelty could be better emphasized.
- Clarify how this study advances knowledge beyond previous research, especially regarding strain diversity or methodological approaches.
- The manuscript requires major language revision.
Specific comments:
Abstract
- The abstract needs some findings to support it more.
Introduction
- Some statements need references such as LAB’s immunoregulatory roles in the first paragraph. Ensure all statements are well referenced.
- Why the selected LAB strains from specific regions (Shanxi and Jiangsu) are of interest?
- The last paragraph of the introduction is being repeated concept wise which was introduced earlier. Consider summarizing these points more briefly.
Materials and Methods
- Please provide some info on the calibration method and quality controls used to validate the measurements.
- What criteria were utilized for selecting the LAB strains (e.g., genetic diversity, source characteristics)?
- Line 88: The phrase "under anaerobic conditions for 2 hours" could be clarified. How those conditions were maintained?
Results
- Include a table summarizing the production capabilities of LAB strains for each indole derivative. This would help readers quickly understand the key findings.
- Visualize geographic or taxonomic differences (if any) with bar graphs or heatmaps.
- The claim that indole-derivative-producing capabilities are "not related to geographical origins" requires statistical support. Provide details of the analysis used to confirm this conclusion.
Discussion
- Discuss how your findings align or contrast with previous studies, such as those cited.
- What are the practical implications of your findings?
- Acknowledge any limitations of the study, such as the focus on specific geographical regions or the exclusion of other indole derivatives.
- Please make sure all figures are clear and include appropriate legends.
Comments on the Quality of English LanguageMajor revisions are needed!
Reviewer 2 Report
Comments and Suggestions for Authors
After some revisions, I believe the Communication submitted by Ma and collaborators can be considered for publication in Microorganisms. These are my suggestions:
“in vitro” should be in italics. Please, check the whole manuscript.
In the abstract, please provide further conclusions from your study and indicate some directions for future investigations. How can the results of your research be impactful and useful?
More keywords should be provided.
Lines 28-29: References are missing.
Lines 70-72: “The results reveal some interesting findings, offering insights into the screening of new functional probiotics and the indole-derivatives metabolizing pathway.” – This shouldn’t be in this section, please delete it.
Please, provide more details about the 16S rDNA sequencing and construction of phytological tree (section 2.2).
The Results section is fine, but the Discussion needs to be expanded and improved. This is a crucial part of the manuscript, and the authors are expected to do more. They need to analyze and discuss the obtained results with the existing literature from a worldwide perspective.
The study’s limitations and strengths should be stated.
Elaborate on your Conclusions, aligned with the abstract you will revise.
Round 2
Reviewer 1 Report
Comments and Suggestions for Authors
Thanks for addressing my comments.
In the introduction, please reorganize the last paragraph so you should have the aim/ objective of the study as the last sentence while the rest of the info (By examining LAB strains from different geographical regions..........) should be before the objectives so you're saying we hope this and that and then mentioning your objectives (in this study, we aim......).
Please make sure to add your supported answers to the papers!
Author Response
Comment 1: In the introduction, please reorganize the last paragraph so you should have the aim/ objective of the study as the last sentence while the rest of the info (By examining LAB strains from different geographical regions..........) should be before the objectives so you're saying we hope this and that and then mentioning your objectives (in this study, we aim......).
Response 1: Thank you for your valuable guidance. We have accordingly reorganized the last paragraph; please see lines 60-66. We also attach the revised paragraph as follows:
In this study, by systematically examining LAB strains from different geographical regions and ecological niches, we aim to identify potential regional, niche, or taxonomic differences in their ability to produce key indole metabolites, including IAId, IPA, IAA, and ILA. Understanding these variations may reveal new insights into how LAB contribute to host health, particularly in terms of immune modulation, gut barrier function, and inflammation, and also provide broader implications for the development of LAB-based functional probiotics.
We hope the current revision meets your standards. Finally, we would like to express our great appreciation for your kind and careful revision of our manuscript. The quality has been greatly improved. Thank you again for the time and effort you have dedicated to our manuscript.
Reviewer 2 Report
Comments and Suggestions for Authors
Thank you for following my suggestions.
Author Response
Thanks to your valuable input, our manuscript has been greatly improved.